# Evaluating Efficacy of a COVID-19 Alternative Care Site Preparedness Assessment Tool for Catastrophic Healthcare Surge Capacity during Pandemic Response

**DOI:** 10.3390/healthcare11030324

**Published:** 2023-01-21

**Authors:** Molly Scanlon, Ellen Taylor, Kirsten Waltz

**Affiliations:** 1Department of Community, Environment, and Policy, Mel and Enid Zuckerman College of Public Health, University of Arizona, Tucson, AZ 85724, USA; 2Research, The Center for Health Design, Concord, CA 94520, USA; 3Architecture & Planning, Johns Hopkins Health System, Baltimore, MD 21201, USA

**Keywords:** alternative care site, built environment, catastrophic events, COVID-19, emergency preparedness, disaster risk management, healthcare design, healthcare surge capacity, pandemic response, public health policy

## Abstract

During the COVID-19 pandemic, implementing catastrophic healthcare surge capacity required a network of facility infrastructure beyond the immediate hospital to triage the rapidly growing numbers of infected individuals and treat emerging disease cases. Providing regional continuity-of-care requires an assessment of buildings for alternative care sites (ACS) to extend healthcare operations into non-healthcare settings. The American Institute of Architects (AIA) appointed a COVID-19 ACS Task Force involving architects, engineers, public health, and healthcare professionals to conduct a charrette (i.e., intensive workshop) to establish guidance during the alert phase of the pandemic. The task force developed an ACS Preparedness Assessment Tool (PAT) for healthcare teams to assist with their rapid evaluation of building sites for establishing healthcare operations in non-healthcare settings. The tool was quickly updated (V2.0) and then translated into multiple languages. Subsequently, the authors of this manuscript reviewed the efficacy of the PAT V2.0 in the context of reported case studies from healthcare teams who developed a COVID-19 ACS in community settings. In summary, policy makers should re-examine the role of the built environment during emergency pandemic response and its impact on patients and health professionals. An updated ACS PAT tool should be established as part of the public health preparedness for implementing catastrophic healthcare surge capacity.

## 1. Introduction

The purpose of this study is to report findings from the coronavirus 2019 (COVID-19) pandemic response for the virus known as severe acute respiratory syndrome coronavirus 2 (SARS-CoV-2) with respect to the integration of the built environment within emergency risk management and public health response models. Specifically, we considered the suitability and design of alternative care sites (ACS) as an integrated solution for catastrophic healthcare surge capacity (HSC) and impact on healthcare professionals. An ACS is a method of developing HSC in which inpatient or outpatient services are moved to an alternate built environment location. This typically happens when (1) the existing healthcare system is overburdened with an immediate hazardous condition (e.g., flooding, earthquake, hurricane) or (2) patient care volumes exceed facility capacity due to emerging disease cases (e.g., outbreak, epidemic, or pandemic) [1,2,3,4,5,6]. The healthcare system essentially needs to establish a field hospital for healthcare operations in non-healthcare settings (e.g., arenas, convention centers, schools, hotels, or other) for emergency or disaster conditions [2]. Optimizing the selection of an appropriate ACS venue involves an analysis of the built environmental conditions [5]. Healthcare building systems are a complex integration of architectural, structural, mechanical, plumbing, electrical, and specialty systems to address functional space allocation and performance, fire and life safety measures, air and water quality, power, data management, security, and other building occupancy standards for patient and staff safety [7]. Licensed design professionals coordinate these building systems using an interdisciplinary team to assure the health, safety, and welfare (HSW) of the public when occupying the structure [8]. During an emergency response, building structures often require assessment and modification to architecture and engineering systems to maintain occupancy to HSW standards [1]. For example, during the COVID-19 pandemic, to reduce transmission in the built environment, modifications were proposed for spatial proportioning for social distancing, cueing lines, physical barriers, and changes in specific finishes and fixtures [9]. Engineering systems were analyzed for increased air exchanges to reduce airborne pathogen concentrations [10], as well as the contributions of toilet flushing activities for pathogen growth and spread [11].

During the COVID-19 pandemic response, licensed and certified healthcare design professionals who were members of the American Institute of Architects (AIA), identified gaps in preparedness [12]. Their initial review of pandemic response guidance documents from agencies such as the World Health Organization (WHO) and the Centers for Disease Control and Prevention (CDC) identified a lack of practical HSW knowledge for catastrophic HSC. As a result, the AIA, a national membership organization for the advancement of the built environment, formulated a COVID-19 ACS Task Force in March 2020 to bring together thought leadership for addressing catastrophic HSC in the context of a global public health crisis [13]. The task force analyzed criteria for implementing an ACS in community settings. The authors of this manuscript, a subset of task force members, conducted a subsequent analysis of the COVID-19 Alternative Care Site (ACS) Preparedness Assessment Tool (PAT) Version (V) 2.0 to evaluate efficacy for future strategic planning for catastrophic HSC during emergency response. The aims of this manuscript were to (1) capture the national AIA COVID-19 ACS Task Force work efforts to formulate the ACS PAT during the pandemic’s alert phase; (2) compare the ACS PAT V2.0 to the published findings of healthcare professionals who mobilized a COVID-19 ACS during the pandemic for evaluating potential efficacy of the tool; and (3) determine improvements for a future ACS PAT V3.0 to benefit interpandemic policy development.

## 2. Background

### 2.1. Emergency Risk Management

Emergencies and disasters utilize a risk management process to reduce human health impacts for any man-made or natural disaster events including a world-wide pandemic response [14]. The WHO states human disease cases, injury, disability, psycho-social dysfunction, and death can be avoided or reduced through implementation of emergency risk management models [15]. Every new catastrophic event reveals gaps for managing health risks. For improving human health outcomes during a response, emergency management traditionally focuses on the health sector and establishing surge capacity as an initial step [16,17]. Increasing numbers of disease cases are managed as the regional outbreak evolves into a national epidemic and continually progresses toward a global pandemic. Implementing the framework of emergency risk management response early in the pandemic cycle can assist the entire community and commerce sectors to meet the wide-ranging needs throughout the lengthy disaster [14,18].

Biological and influenza pandemics are challenging to identify and reoccur at various orders of magnitude across the globe. Influenza A (H1N1 c.2009), the first pandemic of the 21st century [16] and the 2009 Influenza-A pandemic expanded the knowledge of virus transmission. These events also identified the need for effective risk management strategies; exposed the stressful decision making on health agencies; and revealed the challenges of effective public communications. This led to a series of WHO initiatives and guidance documents for risk management of acute worldwide public health events [14,15,16,17,18,19,20].

### 2.2. Pandemic Framework and Phases

A pandemic emergency risk management model involves a global on-going risk assessment to understand the status of the pathogen’s growth and spread [16]. The WHO 2017 Global Influenza Pandemic Risk Management Guidance [16] identified four key pandemic phases—alert, pandemic, transition, and interpandemic. The alert phase is the time when local, national, and global agencies are raising public awareness and assessing if conditions are escalating or deescalating to inform preparedness and taking action. The pandemic phase is the period in which the global spread of the disease is occurring with high human transmission rates. Scientific evidence is used in the fields of virology, epidemiology, and clinical disease and death data to monitor the situation. During the transition phase the global risk will decline, and a de-escalation of activities occurs. However, variants of a virus can mutate and emerge after the initial pandemic phase [21]. These variants tend to spawn outbreaks and regional epidemics that continue until the pathogen’s variant cycle wanes once the population is vaccinated. The interpandemic phase is the time between pandemics for improving future response guidance from the evidence gathered during the pandemic [16]. Movement between phases will be fluid and timing can be rapid or gradual depending upon the continual global risk assessment of the transmission of disease (see Figure 1).

### 2.3. Whole-of-Society Response & Built Environment

A pandemic emergency response requires engagement and activation of all sectors of society to prepare for catastrophic HSC and maintain civil society [18]. The WHO Whole-of-Society (WOS) response model [18] suggested an approach to readiness involves addressing government, civil society, and business sectors. This includes community engagement throughout the pandemic cycle to build trust, share resources, and establish consensus decision making for increased compliance to public health control measures [23]. The WHO’s WOS model states nine essential services: food, water, health, defense, law and order, finance, transportation, telecom, and energy. However, the built environment is not formally mentioned, be it healthcare, schools, housing, governmental, or any other structures. As pandemic preparedness advanced in 2018 the WHO published an updated checklist for building capacity during pandemic response in two key areas [20]. First, identifying facilities appropriate for health and clinical management (i.e., catastrophic HSC), and second, facilities within the community for maintaining functions of civil society (See Figure 2). The WHO’s 2018 [20] updated response identified the need for conducting preparedness assessments of existing facilities and identifying ACS for catastrophic HSC. This step can only realistically be accomplished when considering the time, materials, labor, and logistics for significant changes to the built environment [24]. Yet despite the expertise required, prior to the COVID-19 pandemic, minimal guidance existed about the essential modifications to the built environment for establishing an ACS for catastrophic HSC [6].

### 2.4. Catastrophic Healthcare Surge Capacity

Until the COVID-19 pandemic, catastrophic HSC was largely untested in the international or United States (US) disaster preparedness response efforts. Historically surge capacity has been linked to emergency medicine which deals with high levels of fluctuating census due to crowding conditions [24,25]. However, typical crowding conditions are not normally from a common disease outbreak or even a singular catastrophic event, but rather from patients presenting themselves at an emergency department (ED) who often lack access to a primary care physician and go to a hospital ED instead. Typically, EDs experience daily surge capacity needs which are often a function of cuing challenges and lack of ED and inpatient bed capacity. Common components of daily ED surge capacity include triage, ordering, radiology testing and reading, admissions process, minimal staffing, and supplies [24]. On the other end of the spectrum is the impact of catastrophic HSC related to a disaster event which requires establishing high quantities of additional bed capacity [24,25]. Catastrophic HSC requires a complex integration of resources beyond the walls of the ED and potentially even the hospital campus. The components of catastrophic HSC involve integration of four key elements (4S’s): system, space, staff, and supplies [24,25]. Kelen and McCarthy [24] in developing definitions for ‘The Science of Surge’ mentioned that most published documents lack relevant guidance for establishing these four critical domains which they broadly defined as:System: Components involve planning with community, government, informal networks, public health, regional health systems, hospital epidemiology and infection control, incident command, and local utility infrastructure. Additional systems preparedness involves anticipating supply chain disruption, coordination of first responders, and maintaining continuity of operations including cybersecurity;Space: Catastrophic functional space programming includes determining size and service volumes for medical care, storage, laboratory, mortuary, and staff housing.Staff: Analysis of staffing models for patient care ratios, shifts or rotations, capability and skill sets, expertise, stamina, and psychological impacts; andSupplies: Catastrophic supplies necessitate review of biologics, respirators, personal protective equipment (PPE), and standard use supplies, as well as food and water.

These critical domains were reiterated in the development of a standardized all-hazard disaster training program detailing core competencies (e.g., nomenclature, incident command structures, resource management, go-no-go response teams, and disaster triage, among others) for clinicians and related supporting team members [26]. The Cleveland Clinic described implementing an all-hazard preparedness approach for COVID-19 to establish safe patient care operations during a catastrophic event [27]. Infection prevention and control protocols are also referenced in preparedness as necessary during outbreaks with the notion that lessons learned could be applied to other health threats [23,28]. Even though an ACS environment is noted as part of all-hazard training for catastrophic HSC [26], there is no detail about how to evaluate, select, or modify building systems for a safe and supportive environment of care. Simply mentioning the need to develop plans for catastrophic HSC with no further definition of what this actually entails or the implementation challenges for such a complex endeavor during large scale events puts unnecessary resource demands on an already stained emergency condition [24].

### 2.5. Alternative Care Sites

An ACS is an important option for catastrophic HSC after acute care hospitals have maximized their capacity and capabilities. A wide-ranging term, ACS is intended to define expansion of healthcare facility operations into temporary structures and administer care for a defined patient acuity level [6,29]. Even with national or regional guidance on ACS creation, it is always necessary to adapt an ACS to the specific disaster scenario and the available resources in the local community [29,30]. An ACS can be developed to handle hospital overflow, patient isolation, expanded ambulatory care, recovering non-infectious patients, primary triage, rapid patient screening, or quarantine [2,3,31,32,33,34,35]. Chen et al. [2] described an ACS strategic plan for Wuhan, China with a series of Fangcang shelter hospitals specifically implemented to address the COVID-19 pandemic response. The Fangcang concept was borrowed from military field hospitals yet considered novel to convert large venues into temporary healthcare operations to isolate patients with mild to moderate symptoms of an infectious disease. In Wuhan, China, sixteen buildings were converted into ACS centers adding 12,800 beds within an urban region. Each Fangcang field hospital was intended to reduce community spread and transmission between family members, while providing food, shelter, and social activities. To assist other nations China translated policies and clinical guidelines for international implementation for rapidly growing COVID-19 outbreaks. China sent a delegation of experts with experience in construction and operations of ACS facilities as a form of consultancy to national and local governments such as Italy, Iran, and Serbia [2].

### 2.6. US COVID-19 Developing Situation

The US Federal Administration invoked emergency powers utilizing four statutes (the Public Health Service Act, the Stafford Act, the National Emergencies Act, and the Defense Production Act) for the COVID-19 response starting 31 January 2020 [36]. The US Federal Emergency Management Agency (FEMA) was named as the lead agency in the COVID-19 emergency response efforts. These Acts collectively positioned the US government to provide financial and physical resources to state and local governments to reduce the likelihood of spread and growth of the virus as well as protect the US economy against the pandemic’s escalating effects. As of 23 March 2020, the SARS-CoV-2 virus had transmitted world-wide developing into 369,776 (US, 94,879) disease cases and had resulted in 20,765 (US, 1732) deaths [22].

Many US organizations and governmental leaders initially suggested the COVID-19 pandemic was a ‘black swan’ event beyond emergency preparedness [37]. Introducing statements of uncertainty during a pandemic can create fear, mistrust, and panic [38]. The absence of clarity and trust can lead to a lack of community engagement and ultimately dissuade persons from compliance with public health measures at the local level [38,39]. To promote effective engagement, it is necessary to identify sources of reliable information and avoid spreading rumors [27]. Yet, international emergency and pandemic preparedness documents were widely available to implement some level of preliminary response to reduce illness, injury, and death [14,15,16,17,18,19,20]. However, there was no uniform guidance on disaster preparedness for catastrophic HSC to manage a large rapid human health outbreak, epidemic, or a global pandemic [6,40,41]. Members of the task force, healthcare architects and allied health professionals, were in a unique position to leverage problem-solving knowledge, skills, and abilities to optimize patient and worker safety around built environment modifications [42]. The aim was to establish a checklist for reviewing catastrophic HSC criteria for selecting an ACS building to assure the HSW of all building occupants in the context of launching healthcare operations in non-healthcare building settings. This manuscript describes the process and outcomes of the task force (from 2020), as well as comparing the findings from case studies published (in 2021 and 2022) about ACS implementation during the COVID-19 pandemic response. Our core research question was what criteria would be essential during a catastrophic health event to select and develop a safe built environmental setting for COVID-19 ACS patient care operations? The research objectives were to (1) describe the establishment of the ACS PAT checklist; (2) verify the ACS PAT checklist against reported COVID-19 ACS case studies; and (3) identify knowledge gaps for improvement prior to further dissemination of the ACS PAT checklist among public health agencies and healthcare organizations concerned with catastrophic HSC implementation.

## 3. Materials and Methods

### 3.1. Identifying the Task Force Members

The AIA COVID-19 ACS Task Force was formulated on 19 March 2020 and consisted of members from AIA Academy of Architecture for Health, AIA Design and Health Leadership Group, the American College of Healthcare Architects, the Center for Health Design, the AIA Board of Directors, and the Facility Guidelines Institute (FGI) Health Guidelines Revision Committee [12]. The task force included 12 core members and extended to an additional 40 professionals representing the disciplines of architecture, engineering, nursing, medicine, public health, environmental health science, healthcare codes and standards, and building fire and life safety.

### 3.2. Intensive Workshop—The Charrette Process

The task force used a modified version of an interdisciplinary design charrette. Design charrettes are recommended for interdisciplinary teams in order to expand knowledge from multiple methods of analysis and visualization tools to leverage novel solutions for implementation [43]. During the pandemic, stay-at-home orders were initiated. The AIA task force consequently could not conduct any workshop meetings in person. Therefore, charrette methods were transferred into electronic and digital formats to create a virtual interdisciplinary working environment. Using an online meeting platform with screen sharing, web-cams, digital whiteboards, and chat functions, the task force set goals to (1) review the built environment as a form of hazard control for reducing the likelihood of SARS-CoV-2 virus growth and spread, as well as (2) consideration for managing COVID-19 disease cases in an ACS setting. The task force identified the primary deliverable as a COVID-19 ACS PAT checklist to establish criteria for healthcare teams to evaluate local facility conditions to select an appropriate building site for catastrophic HSC. The task force agreed to meet for seven consecutive days to generate preliminary work products.

### 3.3. Use of Existing Guidance Documents

In addition to their professional experience (e.g., addressing weather-related surge events), the task force utilized the 8-page template for US Army Corps of Engineers (ACE) ACS 250-bed Implementation Support Materials dated 22 March 2020 [44], the 2018 FGI Guidelines for the Design and Construction of Hospitals [41], emerging and published COVID-19 evidence (e.g., Chen et al. [2]), and formats from WHO [15,16,17,18,19,20,22] and CDC influenza pandemic response checklists [45]. The building types mentioned for ACS conversion were hotels, college dormitories, arenas, and convention centers.

### 3.4. COVID-19 ACS Case Study Article Search Method

Using a researched-based university library database system an article search was performed to find ACS case studies addressing catastrophic HSC for COVID-19 pandemic response. The categorical settings from the library’s advanced search engine were used. Search terms were inserted for Title Field: “ACS” OR “alternative care site” OR “field hospital” AND Subject contains: COVID-19 or pandemic. The search was conducted in August 2022 with inclusion criteria for publication dates between 2019–2022, English language, and peer-review or scholarly sourced articles. There were no exclusion criteria associated with geographic location.

## 4. Results

### 4.1. COVID-19 Alternative Care Sites Preparedness Assessment Tool

The task force created the COVID-19 ACS PAT to guide key local and regional stakeholders for evaluating built environment conditions for adaptive re-use of healthcare operations in non-healthcare settings. A first version (V1.0) work product was released 6 April 2020 [46]. An updated version (V2.0) was published on 22 April 2020 [47]. The purpose of the COVID-19 ACS PAT V2.0 was to allow all US states and territories to prepare for the arrival of patients with suspected or confirmed COVID-19 disease cases. The tool contained programmatic architectural and engineering evaluation information synthesized from non-crisis situations (e.g., healthcare design criteria, best practices, available supporting evidence, and applicable healthcare codes and standards). The COVID-19 ACS PAT V2.0 did not describe mandatory requirements. Rather, it suggested a local cross-disciplinary team would strategically evaluate the ACS facility and consider: (1) go/no-go building selection criteria; (2) general conditions and baseline operating parameters; (3) functional program requirements; (4) facility modifications and building infrastructure; and (5) vulnerable populations such as mental health and rural populations. For COVID-19 ACS PAT V2.0 documents see Appendix A for four language versions (i.e., English, French, Spanish, and Portuguese). An outline summary of the COVID-19 ACS PAT is illustrated in Figure 3.

### 4.2. Public Health Dissemination

The AIA created press releases announcing the availability of the COVID-19 ACS PAT V2.0. Various task force members participated in interviews with the national press to disseminate information to the profession, allied public health and healthcare professionals, and the public-at-large. Interviews on the built environment and pandemic response were given to public news outlets [48,49]. Additionally, continuing education sessions were given about risk factors within the built environment that contribute to disease transmission [50,51]. Within 35 days of dissemination (on 11 May 2020), the COVID-19 ACS PAT V2.0 work product was translated by the US Department of State into three languages (French, Spanish, and Portuguese) and disseminated to all US Embassies as support for US citizens working abroad for localized pandemic response [47]. The AIA national website posted these materials in the open domain [12]. From late March 2020 through October 2020 the AIA COVID-19 ACS Task Force materials generated 4085 unique page views from multiple sources (38% within aia.org; 23% from referrals; 23% from organic search engine retrievals, and 10% from email linkages).

### 4.3. COVID-19 ACS Case Study Article Search and Coding Results

The article search method returned 381 articles. An abstract and title review was conducted. Articles were excluded from further review based on the following: (1) acronym ‘ACS’ used for other meanings (e.g., acute coronary syndrome) (*n* = 319); (2) cohort disease studies of patient or staff within an ACS (*n* = 21); (3) general ACS public health response topics (*n* = 8); (4) sub-departmental ACS topic (e.g., pharmacy, laboratory, cost modeling) (*n* = 9); and (5) duplicate article (*n* = 1). The article reviews yielded case studies (*n* = 23) concerning catastrophic HSC for mobilization and operation of a COVID-19 ACS during the pandemic. The articles were read, coded, and compared to the COVID-19 ACS PAT V2.0 five key sections. The key section results indicated alignment and potential efficacy with reported challenges from ACS case studies to set up and operate a safe patient care environment. In particular these were:0.0 Go/no-go selection (*n* = 7, 30.4%);1.0 Baseline operating parameters (*n* = 22, 95.7%);2.0 Functional program requirements (*n* = 20, 87.0%);3.0 Facility modifications (*n* = 14, 60.9%); and4.0 Vulnerable populations (*n* = 9, 39.1%)

An additional section 5.0 ‘other’ was created in response to coding which captured items new to the ACS PAT tool, such as local population challenges related to cultural competency (*n* = 5, 21.7%).

Similarly, each article was scored for ACS PAT V2.0 subtopics numbered 1 through 15. Additional numbers (i.e., 16, 17) were added for subtopics not previously included (e.g., nutrition and language barriers). Summary statistics were performed to record frequency and percentage of ‘yes’ responses for any ACS case study article identifying approaches in alignment with ACS PAT V2.0 criteria. Next, ‘did not report’ (DNR) coding signified the ACS case study article did not provide an explicit description for that topic when comparing it with ACS PAT V2.0 criteria. The ACS facility operations occurred in sports venues [2,29,34,52], convention facilities [30,32,53,54,55,56,57], hotels [31,35], a defunct newspaper plant [58], and a newly constructed yet non-occupied healthcare facility [59], or other new or existing large open space venues [1,3,33,60,61,62,63,64]. Results for each ACS case study article response compared with COVID-19 ACS PAT V2.0 by section and subtopic were summarized in Figure 4. Summary statistics (i.e., frequency and percentage) demonstrating alignment between ACS PAT V2.0 and reported ACS case studies were reported by section and subtopic in Figure 5.

## 5. Discussion

The AIA COVID-19 ACS Task Force developed a tool as a strategic planning checklist for worldwide distribution for establishing healthcare operations in non-healthcare settings. The task force could not control the distribution of the tool or usage once in the open domain. Therefore, comparing and contrasting the COVID-19 PAT V2.0 with peer-reviewed COVID-19 ACS was performed to establish efficacy of the tool moving forward. Twenty-three COVID-19 ACS case studies were reported in US [3,29,30,31,32,34,52,53,55,56,57,58,59,61] and International [1,2,33,35,54,60,62,63,64] settings. Our discussion illuminates findings and areas for improvement.

### 5.1. Go-No-Go Building Selection

The COVID-19 ACS PAT V2.0 recommended a ‘Go-No-Go’ building evaluation to identify complications with ASC facility modifications and potentially dangerous HSW concerns (e.g., building age, life-safety, and available utilities) prior to building occupancy. The ACS case study articles (*n* = 7, 30.4%) reported alignment with ‘Go-No-Go’ criteria. Even though federal or state/province agencies evaluated building criteria for pandemic response, local clinical teams were faced with challenging patient care environments [3,55,58]. For example, the State of California declared a public health emergency to commandeer and repurpose buildings such as hotels, gymnasiums, or previously shuttered healthcare facilities [3]. Although each facility could accommodate space for patient bed configurations, Christensen et al. [3] reported numerous building infrastructure challenges concerning patient and staff safety were not addressed. Environmental concerns surfaced over poor water quality; lack of adequate heating, cooling, and electrical power; and an inability to maintain patient oxygen concentrations. Power outages commonly occurred, which required safety officers designated at each site to monitor for electrical circuit overloading and potential fires.

Similarly, Mathews et al. [59] described a case in which the New York State Government commandeered a newly constructed 216-bed psychiatric facility in Staten Island for use as a COVID-19 ACS. Although the site was a brand-new facility, the building had to undergo renovation for COVID-19 patient care. Hospital psychiatric room doors were a reduced size and could not accommodate an acute care hospital bed with rails. Additionally, the psychiatric facility was designed with communal restrooms which were not ideal for reducing infectious disease transmission. A lack of electrical outlets also proved problematic since every psychiatric patient room had only one outlet as a standard safety precaution. Other deficient building systems for acute patient care operations were limited internet infrastructure for medical record documentation, medication storage, PPE storage, and high-volume medical supplies. Lastly, there was no on-site kitchen for dietary preparation other than food warmers, which is common for psychiatric facilities utilizing a contracted food service vendor.

Furthermore, a hectic building selection and uncoordinated response can create dangerous HSW conditions and use scarce resources unnecessarily. During the initial phase of COVID-19 response, patients were triaged in parking lots and tent structures [3,33]. In Shanghai, China, a hotel was used as a COVID-19 quarantine facility for citizens exposed to the virus [65]. The seven-story steel structure hotel collapsed trapping 71 people and resulting in 10 deaths. A regional government deployed over 1000 fire fighters and first responders to the collapsed hotel site in a rescue mission to reach survivors. These ACS examples demonstrate the need for criteria to evaluate a ‘go-no-go’ building selection. Not every facility will be ideal, and infrastructure may need adjustments, but a building preparedness assessment tool used in advance of mobilization would likely improve effectiveness of an ACS operations launch.

### 5.2. General Conditions

#### 5.2.1. Baseline Operating Parameters

Baseline operating parameters were the most commonly reported criteria in alignment with the ACS PAT checklist. Twenty-two (*n* = 22, 95.7%) ACS case studies reported at least one baseline parameter for managing ACS operations. The top three baseline operating parameters reported were establishing a healthcare operating authority (*n* = 15, 65.2%); developing a staffing model (*n* = 13, 56.5%); and implementing an infection control donning and doffing method for controlling PPE (*n* = 11, 47.8%).

##### Establish a Healthcare Operating Authority

Under the general conditions section, the COVID-19 ACS PAT V2.0 stated an operating authority having jurisdiction (AHJ) must be identified. Both US and international case studies reported collaboration between national and local agencies, as well as public health and healthcare delivery system coordination to establish a functioning ACS for pandemic response [2,3,58]. Once the ACS was established, healthcare teams described an incident command center system of management to achieve universal communication under complex conditions [3,29,31,53,55] modeled after military emergency response operations.

In Boston, Massachusetts, local government and healthcare systems cooperated to designate the convention center as a 1000 bed ACS [53]. The first 500 beds were under the direction of a local homeless healthcare agency for undomiciled citizens with COVID-19 requiring isolation. The second 500 beds were post-acute care patients under the jurisdiction of a non-profit multi-healthcare system consisting of two academic medical centers, a post-acute care patient network, and the local community hospitals. The AHJ was responsible for financial, operational, and human resource allocation to manage the ACS. The AHJ’s incident command center key leadership consisted of two co-directors, a military task force army control officer, a chief medical officer (CMO), chief nursing officer (CNO), and chief of clinical operations.

In Memphis, Tennessee, a shuttered newspaper publishing building was converted into a 402-bed ACS COVID-19 transition care center for additional time for patient recovery prior to returning home [58]. The Memphis ACS engaged the state governor’s office, state department of health, county government, US FEMA, and state emergency management. Additional key team members for implementation and communication were the local media and the construction company to renovate the facility. Leadership was accomplished using a CNO model with knowledge of regulations, infection control, hospital design, and support services. A succession plan was deemed important to create leadership redundancy due to virus transmission or the need to quarantine healthcare leadership for extended periods of time.

##### Safety Risk Assessment & Running Simulation Exercises

A Safety Risk Assessment has become a standard healthcare facility planning tool in the US [66,67,68], and is part of the most current edition of the US-based FGI *Guidelines* including a new component for Disaster, Emergency, and Vulnerability Assessment [7]. Although the COVID-19 ACS PAT V2.0 stated a need for a safety risk assessment, the ACS case studies (*n* = 6, 26.0%) recognized the supplemental need to run various simulation exercises to assure safe healthcare operations in non-healthcare settings [1,52,55,58]. These simulation exercises (e.g., tabletop and daylong operations) led to rearranging the physical environment prior to admitting patients [1,58]. The tabletop exercises were discussions reviewing facility layout for leadership, management, staff orientation, staff safety, patient management, and external service coordination to medical centers [58]. During the second phase, daylong exercises tested and evaluated operations simulating patient care flow from admissions to discharge. Gaps were identified and changes implemented. Additionally, drills were run to stress test the conditions and determine if the healthcare teams understood when to seek resources or leadership to assist in healthcare operations problem solving.

##### Infection Prevention & Control

The COVID-19 ACS PAT V2.0 tool, similar to the reported ACS case studies emphasized proper infection prevention and control (IPC) practices as a critical focus of the ACS facility conversion and healthcare operations process. However, the ACS case studies (*n* = 7, 30.4%) emphasized a tri-level hierarchy of infection control zones to strategically utilize PPE in contaminated areas (high risk), semi-clean areas (medium risk), and clean zones (low risk) [2,32,52,55,57,59,60]. The ACS IPC methods described functional space designations of high-risk patient care areas, medium-risk clinical staff areas including donning and doffing PPE, and low-level areas for administration and support staff where minimal PPE would be worn. The Philadelphia ACS chose an IPC organizational practice utilizing colored zones (e.g., red, yellow, and green) to limit viral spread among ACS building occupants [52]. Red zones were for high monitoring activities and restrictive access near patient care and decontamination activities. Yellow zones were for staff donning and doffing of PPE and acted as the interface between red and green zones. Green zones were designated for public access and neutral hallways near entrances and exits or staff circulation zones for which only a basic surgical mask was required. Due to the high demand, it was challenging to recruit IPC practitioners. To compensate, the Philadelphia ACS secured public health nurses and general nurses with IPC knowledge. The team developed IPC designees who were trained and coached to enforce safety and IPC standards of operation.

A key aspect of ACS facility IPC protocols (*n* = 11, 47.8%) involved developing donning and doffing stations and hygiene facilities [3,31,32,52,53,54,55,57,58,59,60]. Donning and doffing stations in simple terms are the changing areas for putting on (clean) and taking off (contaminated) PPE. These spaces were frequently assigned IPC monitoring staff to deal with observing PPE protocols; updating staff for changes in PPE protocols; or training to appropriately wear PPE (test and fit) [3,52,53,55,57,60]. Adherence to strict 24-h monitoring was implemented to avoid self-contamination and infecting large numbers of healthcare staff in a crisis [55]. Jones et al. [57] described bringing a hygiene trailer to the ACS facility for staff to shower and change into clean clothes upon exit to allow a safer transition to their home. Similarly, the 850 US Public Health Service (PHS) Corps officers at the New York City (NYC) Javits Center with 2500-beds were in charge of the one-way flow for IPC standards at the facility and staff training [55]. Many US PHS officers were drawing experience from operating an Ebola treatment center in Liberia, West Africa in 2014. Patient care areas were segregated from non-patient care areas to differentiate PPE use. Checklists and scripts were written for donning and doffing as well as posting sign-in sheets at patient care areas to monitor PPE usage, avoid self-contamination, promote consistency, and establish PPE “burn rates” to manage inventory.

Although the COVID-19 ACS PAT mentioned IPC practices and establishing donning and doffing stations, an updated checklist of spaces, methods, and engineering controls should be considered based on the ACS case study review. The IPC practices were frequently mentioned (*n* = 12, 52.2%) as key space planning aspects impacting overall facility selection, design, and evolving healthcare operations in the early stages of the pandemic response.

##### Collaborating with Healthcare Design Professionals

The COVID-19 ACS PAT V2.0 suggested engaging licensed, trained, and experienced healthcare architects and engineers familiar with catastrophic HSC for ACS building conversions. The ACS case studies reported a variety of physical facility conditions suggesting the facility was set up by others and subsequently the local team was left to manage the situation with minimal or no support [3,58]. Only two (*n* = 2, 8.7%) ACS case study articles reported collaborating across the pandemic cycle with healthcare design professionals. The US FEMA ACE 250-bed template facilities were reported to have been constructed in 37 cities across the 50 states using $660 million of US federal resources [6,32,44,58]. International ACS development reported using WHO ACS guidelines [1,60] and China distributed the Fangcang shelter hospital concepts for worldwide use [2]. There were a wide range of experiences using these ACS facility toolkits—most of which were not written or available prior to the COVID-19 pandemic. Following an initial vision of constructing 1000 non-COVID-19 beds, which was adjusted within days to 2500 beds for COVID-19 care, the NYC Javits Center [55] which obtained US ACE ACS Implementation Support Materials successfully constructed a 48-bed ICU to treat COVID-19 patients during this crisis. Meanwhile, Stewart and colleagues [58] in Memphis, Tennessee described a painful process of transforming a defunct newspaper plant into an ACS. Although US federal agencies provided the 250-bed template and site materials, the Memphis ACS implementation was a state-led effort. Five weeks were lost to facility adaptation to flex up to a 402-bed facility during a critical time window which could have been used to treat more COVID-19 patients to reduce illness, injury, and death. Stewart et al. [58] reported having no credible information on converting a commercial building into a health center, as well as the stressful working conditions involved in establishing a proper nursing standard of care in a limited time period. Fortunately, the Memphis ACS benefited from a savvy and experienced CNO who developed most of the functional planning requirements for nursing stations, medication rooms, clean and dirty supply rooms, nutrition, and hygiene stations.

In contrast, Castro-Delgado et al. [1] described a multi-disciplinary approach in Asturias, Spain in which healthcare design and construction professionals were imbedded into the ACS implementation team. The ACS implementation team described architecture, engineering, nursing, and medicine as integral partners working throughout the pandemic response not only to design the initial facility layout but to remain involved to adjust the facility over time. Running simulation exercises, changing medical equipment, and evolving patient treatment scenarios all dictated changes to the physical environment that required technically healthcare-competent architectural and engineering staff to perform. These healthcare design professionals were part of the local health department and incorporated into the pandemic response. The authors reported the physical environmental changes led to team building exercises which in turn stimulated better communication and effective implementation of new protocols over time.

Healthcare facility planning, design, and construction professionals have specialized in guidelines, standards, and codes for the HSW of patients, visitors, and staff dating back to the implementation of the US Hill Burton Act of 1947 [69]. Today, licensed and certified healthcare architects and engineers are available for consultation in many countries [42]. Key non-profit organizations exist for the professional guidance, consultation, and research associated with healthcare facility regulations, best practice, and evidence-based design [13,70,71]. There is no need for any organization or governmental agency to struggle with determining how to site adapt regulatory guidance for catastrophic HSC for any local, regional, or national disaster. Yet, these same healthcare design professionals likely need all-hazard disaster core competency training [26] similar to healthcare professionals to understand the differences between standard healthcare operations and catastrophic healthcare operations (e.g., nomenclature, disaster triage, incident command structure, communication, and record keeping).

#### 5.2.2. Temporary Assets

##### Staffing

Within the ACS case study reports (*n* = 13, 56.5%), there were varied methods for establishing a temporary healthcare staffing model as well as where to obtain staff. Healthcare leadership and staff were recruited from contracted health staffing agencies [30,52,59], local healthcare organizations [53,55], nursing or medical schools [3,53], staff on furlough from suspended health services [33], the US PHS Corps [3,55], Ebola teams from prior infectious disease outbreaks [55,60], military and reserve service [3,53,55,63], paramedics [29], state emergency medical teams [3,29,52], vendors [52], homeless shelter staff [61], and volunteers with related skills [3,61]. The wide variety of clinical staffing led the Colorado Department of Health to create COVID-19 clinical training modules to increase team performance under various environmental conditions [72].

##### Medical Equipment

An ACS environment had to obtain medical equipment, supplies and PPE to operationalize a safe environment for patients and staff. Finding or procuring large quantities of items proved to be very challenging [34,63]. Bell et al. [34] reported even basic essential supplies were difficult to obtain when the entire US was in a rapid response mode. Their team struggled to find cots, linens, and privacy screens. Similarly to the grocery shelves running bare during a disaster, healthcare items were reportedly out of stock [34]. The competition for resources between local, state, and national stock piles were also exhausted [6]. Additionally, wi-fi network access was critically important to link to medical equipment and establishing an electronic medical record system. The Memphis, Tennessee ACS site chose paper charting methods due to lack of trained digital network staff and resources to implement [58]. Baughman et al. [53] leaned on their multicentered healthcare system approach to acquire medical equipment, supplies, and establish the electronic medical record system. In contrast, an ASC established in a hotel did not have any traditional medical equipment, systems, or supplies [31]. The clinical team cleverly used existing equipment and systems for medical purposes. Kaysin et al. [31] described leveraging refrigeration for personal medications, safe storage for belongings, on-site food services, security systems to control access, and wi-fi capabilities for connecting to local electronic medical record systems.

#### 5.2.3. Evolving Recommendations

Even after ACS site adaptation was completed and patients were admitted, the ACS case studies reported the need to continually reassess environmental conditions and make adjustments to address patient treatment issues. Baughman et al. [53] described a convention center with 500 post-acute care cubicles, yet bathroom access initially was the convention center’s typical multi-station restrooms located some distance from the patient care areas. Weak patients, as well as dementia patients could not walk the distance to go to the restroom or navigate the large-scale environment. To remedy the situation, handicap accessible bathrooms were erected closer to patient care areas. The ACS facilities for New Orleans, Louisiana [30] and Asturias, Spain [1] had difficulty with COVID-19 patients who developed a form of patient psychosis (i.e., disorientation with regard to time, space, and place) due to being in a convention hall or exhibition space with no natural light. In response, Castro-Delgado et al. [1] described the ‘humanization plan’ to reduce fear for patients and staff. This included spaces for reception and information areas for relatives of admitted patients; areas for patients to walk outdoors with staff when medically mobile; and a designated family visitation room to deal with terminally ill patients. Maslanka et al. [30] described similar challenges managing an ACS facility with elderly nursing home patients. The healthcare team developed a patient engagement program to reduce patients’ cognitive decline. The patient engagement program included structured activities such as chair yoga, religious readings, and stretching. They also brought in televisions, tablets, games, coloring books, and craft supplies. A similar wellness program was implemented at ACS Boston [53] which implemented yoga, mindfulness, bingo, and visual arts. Digital tablets were used to supplement information on exercise, nutrition, spirituality, music, and other entertainment content. Staff reported these efforts resulted in an immediate mood improvement among the patients [30].

After patient care operations had commenced, the ACS case studies also described the need for on-going protocol adjustments to manage unforeseeable operation challenges during a pandemic response [1,32,55]. Thompson et al. [55] described implementing safety officers as liaisons with staff to identify non-compliant protocols and if necessary, suggest a revised approach. Staff were notified of protocol updates at the change of each shift and through chain of command communication strategies. Chaudhary and colleagues [32] established a Rapid Response Team (RRT) to deal with deteriorating COVID-19 patients who required a treatment intervention to then subsequently transport patients to a higher acuity healthcare setting. The RRT team developed an emergency drill process which identified challenges with functional space layout, medical equipment storage, lack of overhead paging, and staff unfamiliar with the role of RRT teams. Solutions were implemented to resolve each challenge.

### 5.3. Functional Program Requirements & Concepts for Operations

#### 5.3.1. Identify Admissions Criteria

From the beginning of the COVID-19 pandemic there was continual confusion about how an ACS could seamlessly operate and complement the local existing healthcare systems in terms of catastrophic HSC. A majority of ACS case studies (*n* = 14, 60.9%) described specific patient admissions and transfer criteria [1,2,3,29,30,31,33,34,35,53,54,59,61,64]. The ACS case studies wanted to ensure that patient admissions occurred with adequate: (1) clinical treatment methods; (2) infrastructure, equipment and supplies; and (3) if patient outcomes escalated beyond the facility’s resources a clear transfer policy was in place.

#### 5.3.2. Site Selection and Location

The COVID-19 ACS PAT V2.0 identified space and planning criteria to establish a safe environment for patient care operations. This ranged from appropriate site selection to ensuring the space could accommodate the defined number of building occupants. Yet, four ACS case studies [29,33,52,57] implied that they started with nothing more than the broad category definitions of surge capacity known as the 4S’s—structure, stuff, staff, and systems [24,25]. Additionally, the time to execute an ACS site adaptation was unknown. The COVID-19 ACS PAT tool did not address any temporal issues as criteria for site selection and facility conversion. However, both US and international ACS case studies reported venue conversion time as impacting their ability to manage disease cases within an escalating pandemic [2,31,52,58,59,63]. The ACS case studies reported site adaptation ranging from a high of 35 days [58] to a low of 7 days [59] not including China. Implementing the US FEMA ACE 250-bed generic template required considerable ACS site adaptation time to local building conditions. For example, the Philadelphia ACS university gymnasium reduced the 250-bed template to a 152 low-acuity COVID-19 patient bed capacity in 3-weeks in order to maintain 6-foot social distancing and other IPC practices [52]. Yet, China and the Fangcang shelter hospital concept reported converting large scale venues in 1–2 days [2]. The first three Fangcang shelter hospitals in Wuhan were completed in 29 h and established 4000 beds for regional catastrophic HSC.

For site selection in India, a template-based hangar design was used for multiple locations to manage COVID-19 patients [64]. The Delhi team evaluated the site selection using Geographical Information System (GIS) mapping, using criteria for ambulance accessibility, roads to the entrance, proximity to an arterial road and other transport (e.g., rail, air, bus), and separation from residential areas. The site also needed to be located to mitigate the risk of flooding.

### 5.4. Facility Modifications & Building System Infrastructure

The COVID-19 ACS PAT V2.0 described the need to evaluate significant facility modifications to building systems impacting structural, electrical, plumbing, and mechanical systems. COVID-19 ACS case studies (*n* = 14, 60.9%) reported conflicts with major building systems that either required modification or in some cases were never analyzed prior to admitting patients [1,2,3,29,30,52,54,55,57,59,60,62,63,64].

#### 5.4.1. Structural

While structural systems are notably difficult and expensive to adjust, there were locations (*n* = 2, 8.7%) that found the need to make adjustments due to the local conditions. In the case of the purpose-built hangar facility in Delhi, India, the structure was subject to wind-related roof lifting, and the close proximity of the structures resulted in a Bernoulli effect [64]. The team needed to include subsurface pile foundations, as well as the fastening of the structure with stainless steel rope secured to reinforced cement concrete foundations, and roof structure bracing.

#### 5.4.2. Mechanical Air Systems

With the virus spreading from aerosolized droplets, mechanical air flow systems were frequently modified (*n* = 8, 34.8%) to increase air flow and air exchange rates [1,2,3,10,52,55,57,60,63,64]. The NYC Javits Center modified engineering controls, since the patient cubicles were not individually ventilated due to the open-air exhibition hall [55]. The mechanical air systems were changed to establish negative pressure air flow across the open area in an attempt to disperse aerosols. Additionally, outdoor air circulation was increased to improve air quality and not lose thermal comfort. Staff at the facility routinely used the ‘tissue test’ to verify the air flow direction. By holding a tissue overhead, the staff could discern at the patient care area if there was continuous flow of air moving over the cubicle areas. An ACS in Bergamo, Italy modified the mechanical air flow rates in a fair market trade center to six air changes per hour in patient care zones [60]. They installed fans to direct air flow from clean to dirty zones. As an additional hazard control step, the final air exhaust to the outdoors was then passed through ultraviolet lamps before releasing the air into the community.

#### 5.4.3. Electrical Power

A common deficiency was inadequate electrical power (*n* = 6, 26%) [3,29,59,62,63,64]. In Delhi, the electrical design of the ICU included an inadequate number of plugs per bed, which needed to be augmented [64]. There was also no time to test the electrical load until two days before the first patients were admitted, and load testing failed twice resulting in the need to upgrade the system. Breyre et al. [29] in Imperial County, California set up an ACS facility at a community gymnasium. The initial ACS functional needs exceeded the facility infrastructure capacity for electrical power and diesel generators were installed for additional emergency power.

#### 5.4.4. Plumbing

##### Medical Gas Systems

COVID-19 patients were frequently treated with oxygen, yet insufficient medical gas systems within ACS facilities (*n* = 9, 39.1%) were commonly reported [1,3,30,54,55,59,62,63,64]. Although the facility in California [29] was initially designated for stable COVID-19 patients without oxygen needs, providing supplemental oxygen became necessary as symptoms of ‘long COVID’ emerged. Oxygen delivery modalities were a challenge due to overload of normal electrical power, lack of emergency power supply, and difficulty with replenishing oxygen supply tanks. This resulted in searching for a source for oxygen tank replenishment.

##### Potable Water Systems

Poor water quality and inadequate access to hygiene stations (e.g., restrooms or bathrooms) were also concerns (n = 4, 17.4%) in ACS environments [3,52,57,63]. In Baltimore, Jones et al. [57] implemented a robust IPC program inclusive of water safety concerns about *Legionella* transmission. The IPC team realized the convention center had unknown water quality from low occupancy due to stay-at-home orders prior to the ACS conversion. The building was set-up for long term operations during the pandemic (e.g., COVID-19 patients, monoclonal antibody infusions, community virus testing, and vaccination). Due to the rapid ACS facility set up, the IPC team did not have time to formally test the building water system. However, they did implement flushing water through fixtures for all sinks and showers. These controls likely reduced water age and increased water disinfectant residuals for better water quality and safety [73]. During the pandemic additional tools emerged to assist building owners and healthcare teams with managing waterborne pathogens due to low occupancy, shut-downs, change of occupancy, or construction activities [74,75,76].

### 5.5. Considerations for Vulnerable Populations and Cultural Competency

The COVID-19 ACS PAT V2.0 suggested an ACS operation would need to address vulnerable populations in rural settings, ethnic diversity, marginalization, homelessness, communal housing, or persons exhibiting mental and behavior health conditions. None of the ACS case studies specifically addressed rural populations and challenges of public health access to treatment. However, the ACS case studies (*n* = 9, 39.1%) did report managing homelessness, mental and behavior health challenges, as well as cultural competency challenges (*n* = 5, 21.7%) with language and nutritional barriers impacting health status.

#### 5.5.1. Urban Homelessness & Communal Housing Response

The ACS case studies reported managing large populations related to urban homelessness, persons living in poverty, or families in overcrowded housing districts [31,35,61]. In Buenos Aires, Argentina, an urban city of approximately 3.1 million people, an estimated 330,000 people live in slums and shared housing [35]. As part of a large-scale public health response, the City’s Minister of Health commandeered 46 hotels which were closed during the pandemic. Remodeling teams, equipment, and health staff were assigned to each location for handling low-acuity COVID-19 cases [35]. The hotels housed 25,813 people during the pandemic. Sixty-two percent were persons with a positive COVID-19 test while 38% were persons under investigation who had been identified as a potential virus carrier through public health contact tracing. The Ministry of Health’s approach of isolating those COVID-19 cases living in slums and shared housing was believed to have significantly reduced hospital caseloads. Only 5% of the ACS hotel population were transferred for high acuity hospitalization.

#### 5.5.2. Mental & Behavioral Health Response

The ACS case study patient admission criteria frequently excluded or discharged anyone who was presenting with severe behavioral or mental health conditions such as disorientation, dementia, suicidal or homicidal ideation, or personality disorders [2,29,31,59,61]. If a patient, post-admission developed significant changes in mental health status, the designated clinical team was summoned at the facility to determine the level of care, mitigate the situation, and if necessary, initiate a patient transfer [59]. An alternate solution for post-admission behavioral and mental health cases was to perform a telehealth consult [2,3,31]. Telehealth consults allowed for assessment and intervention without having full time staff at the ACS. The ACS Boston team [53] had an on-site mental health team who provided consultation to 25% of the patients admitted to the ACS. The NYC Javits Center ACS team [55] found themselves challenged to offer critical care services to behavior health patients in a field hospital. They suggested these persons were displaced from public city shelters [55] and would recommend handling behavior health patients differently during future health outbreaks.

Taking a different approach with recognizing the complexities of inner-city populations, the Cook County Chicago ACS [61] described a more robust clinical admissions approach to manage behavioral and mental patients needing COVID-19 care. Eighty-eight percent of the patients admitted had a definable mental health condition. The Cook County ACS accepted patients with a substance use disorder, clinically stable disorder appropriate for outpatient treatment, criminal history, or chronic hemodialysis. However, like other centers, they did exclude or discharge patients with severe disorientation, uncontrolled psychosis, or displaying suicidal or homicidal ideation.

#### 5.5.3. Language Barriers

Some ACS case studies reported diverse populations creating cultural issues related to language barriers [3,29,30,53]. A number of persons admitted at regional ACS locations were Spanish speaking [3,29,53]. In addition to the common language translation issues, wearing PPE with high levels of background noise reduced effective communication to these patients [30,53]. To address language barrier issues ACS case study clinical teams recruited Spanish language medical staff [3,53], and translated discharge documents and facility signage [29]. Maslanka et al. [30] at the New Orleans ACS contracted a video interpretation service for both verbal and sign languages to minimize communication barriers with non-English speaking patients.

#### 5.5.4. Nutrition Barriers

ACS case studies reported nutrition barriers related to ethnic foods [3,30]. Some patients had culturally specific dietary needs which were not accommodated in initial meal planning. Therefore, lack of food intake led to weight loss and other health complications with COVID-19. Patients wanted home cooked traditional foods which were not attainable due to limitations of family visitation [3]. To resolve this issue, families were allowed to take traditional meals to the contracted catering company providing food to the ACS facility. The items were reviewed, packaged and transported to the ACS. When patients took advantage of this dietary option, healthcare staff noted improved nutritional intake, better mood, and other positive psychological outcomes. With senior age patients Maslanka et al. [30] reported potential malnutrition and having to make sure proper soft foods were available for those elderly patients with oral hygiene and chewing issues.

### 5.6. Limitations

The AIA COVID-19 Task Force met under challenging conditions in an all-virtual format and produced documents for release over an 18-day period. Although the task force’s pandemic work experience assisted to inform the response, like all other entities during the pandemic, we were caught in a vicious whirlwind of activities reflective of a severe lack of preparedness. Additionally, our subsequent review of COVID-19 ACS case studies was limited to findings published in peer-reviewed journals during the search time period. Future analysis of these and other ACS case studies may review patient or staffing outcomes from treating patients in such environmental settings.

## 6. Conclusions

In spite of all the world’s advancements in global health, medicine, and science, society was incapable of launching a comprehensive emergency risk management approach inclusive of built environment modifications during the COVID-19 pandemic. As of early October 2022, the SARS-CoV-2 virus has transmitted world-wide resulting in 615.3 million (US, 94.9 million) disease cases and 6.5 million (US, 1.0 million) deaths [22]. Although lessons from prior outbreaks and epidemics have informed domestic and international preparedness [16,19,45], the COVID-19 pandemic appeared to overwhelm the existing models [6]. Similarly, with all the advancements in healthcare architecture and the built environment, there was no uniform guidance on catastrophic HSC during a health disaster [40,42]. The US and international public health agencies appeared unprepared to suggest rapid modifications to the built environment for adequate hazard control options. Collectively, public health and healthcare preparedness must change across agencies and professional disciplines to reduce the likelihood of repeating such a poor response to catastrophic HSC. As the COVID-19 pandemic response ebbed and flowed around the world, the need for catastrophic HSC remained. Moving forward the authors would recommend:Review and revise the COVID-19 ACS PAT V2.0 checklist based upon ACS case study findings. Specific areas for improvement within a future ACS PAT V3.0 include:
Healthcare operating authority: identify an incident command center as the common method of operations and allocation for space and communication systems for implementation.Infection prevention and control zones: organize a tri-level hierarchy of infection control zones (e.g., low, medium, high) as a space planning concept which includes varying levels of PPE for donning and doffing procedures at the transition between zones.Safety Risk Assessments/Simulations: patient care and staff operational simulations should be conducted at various stages of ACS development for making adjustments to the built environmental setting as early as possible including preliminary design reviews, tabletop exercises, and pre-admission operational drills. Additionally, anticipate built environment changes throughout the pandemic as the pathogen of interest evolves and medical treatment changes.Functional Program Requirements: each site will need to be continually evaluated and site adaption may be necessary for an evolution of pandemic response that can impact building systems, space allocation, staffing patterns, and medical equipment.Electronic Medical Records: anticipate and determine the need to connect the ACS to a local healthcare organization’s existing method of medical record input and archival. Parallel existing systems to reduce logistical difficulties for accurate documentation from initial patient admission.Electrical Power and Medical Gas Provisions: during respiratory virus transmission oxygen supplies will be impacted. As the COVID-19 pandemic evolved (i.e., concepts of ‘long COVID’ emerged) approaches to admitting, treating, transferring, and discharging patients were impacted by access to building oxygen supplies and subsequently emergency power resources for ventilators and other medical equipment.Cultural Competency: include nutrition and language barriers that may arise in local populations which impeded patient care operations and had potentially negative patient outcomes.Introduce all-hazard disaster preparedness training [26] for design professionals to more effectively participate within a public health or clinical response team in order to better guide organizations through the complexity of built environment modifications for catastrophic HSC. This public health architect and engineer role would anticipate built environment challenges and reduce the burden of ad-hoc facility adjustments by public health or clinical teams during pandemic response.Engage public health and healthcare organizations in catastrophic HSC and disaster preparedness at the federal, state, and local levels to create awareness in understanding the role of the built environment. Encourage community engagement to determine an appropriate ACS facility during the interpandemic phase of response which in-turn will identify human and material resources, community partnerships, conflicts of interest (i.e., bias), and methods of recovery [27].

In conclusion, documenting and disseminating the advantages and disadvantages of the ACS implemented facility solutions is a crucial first step for proactive response under conditions of uncertainty. Hindsight bias may always be an issue in evaluation of what was not successful, yet having a published base of knowledge will advance our collective understanding for future emergency preparedness training and response.

## Figures and Tables

**Figure 1 healthcare-11-00324-f001:**
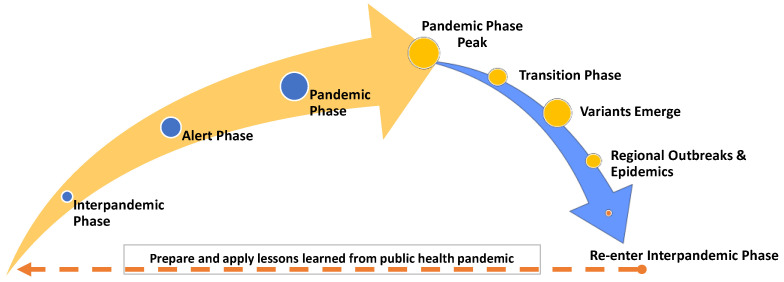
Conceptual framework for emergency risk management during pandemic response [14,15,17,18,19,20,21,22].

**Figure 2 healthcare-11-00324-f002:**
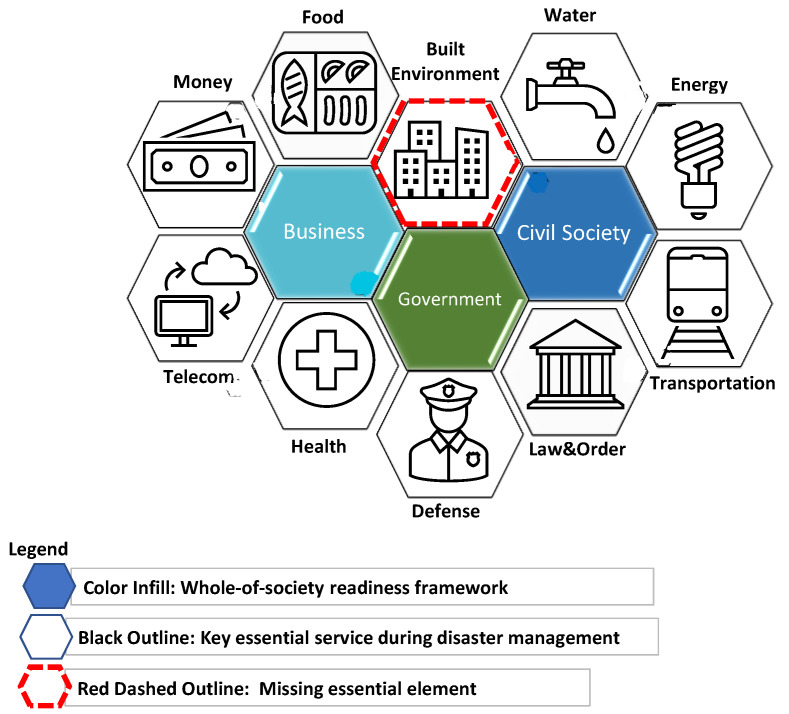
Whole-of-Society response inclusive of the built environment [12,14,18].

**Figure 3 healthcare-11-00324-f003:**
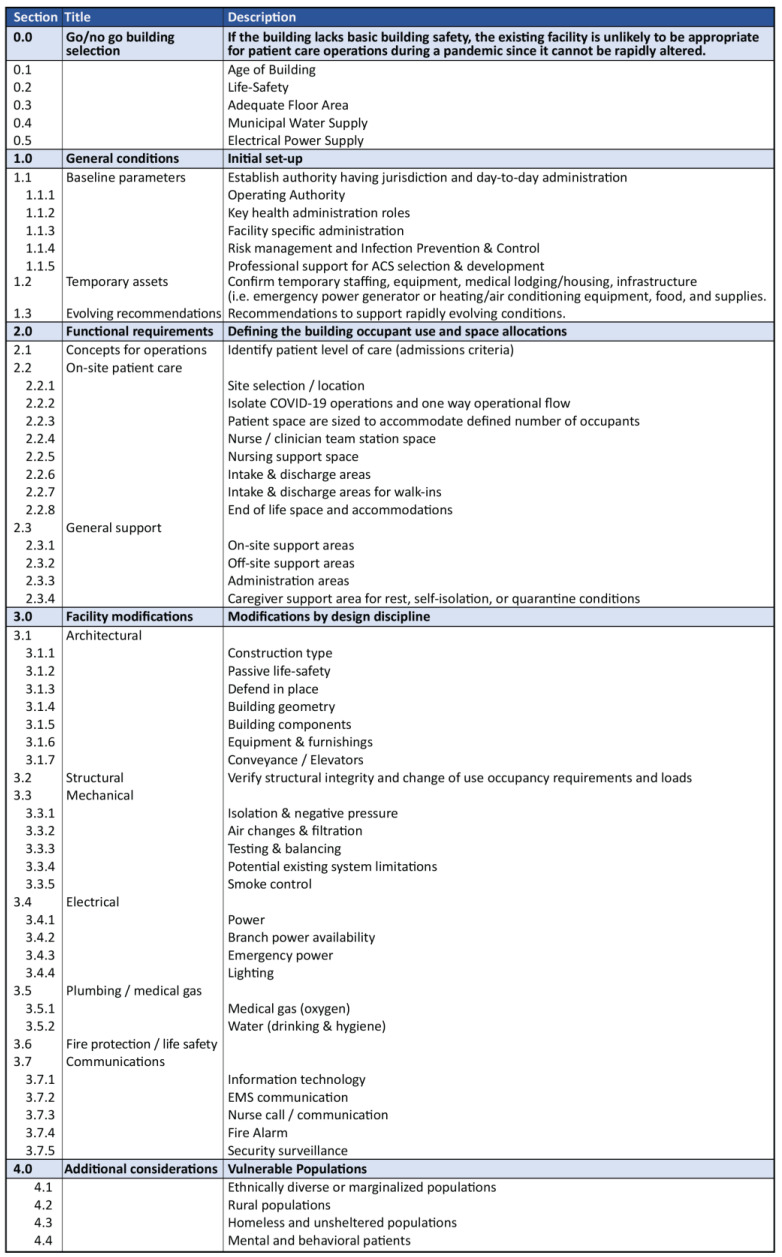
COVID-19 ACS Preparedness Assessment Tool Outline Summary.

**Figure 4 healthcare-11-00324-f004:**
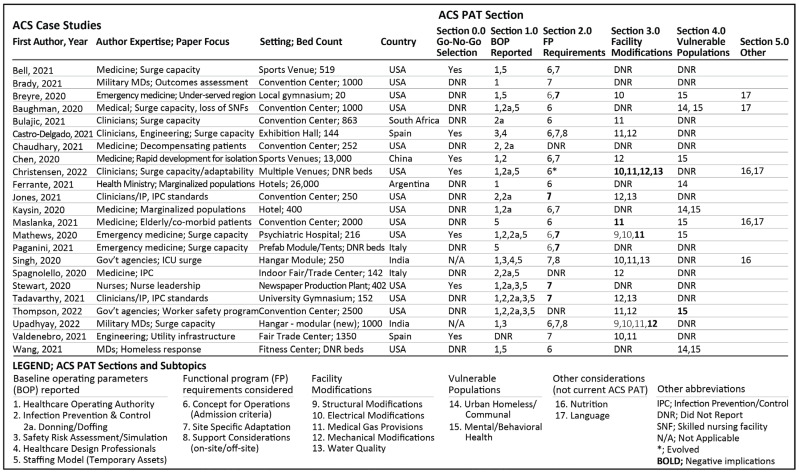
ACS Case Study Summary Comparison with COVID-19 ACS PAT Criteria.

**Figure 5 healthcare-11-00324-f005:**
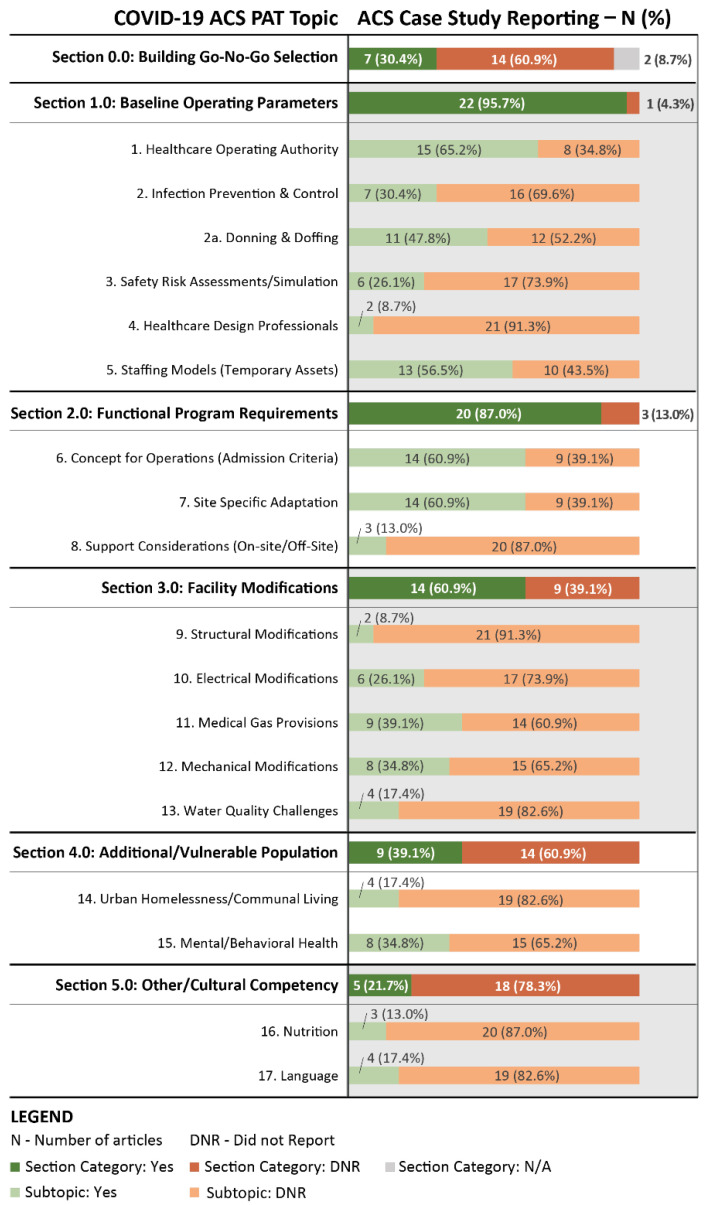
Summary statistics of alignment between COVID-19 ASC PAT and ACS case studies.

## Data Availability

Data is contained within the article or Appendix A.

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
