# Peer review of "Evaluating Efficacy of a COVID-19 Alternative Care Site Preparedness Assessment Tool for Catastrophic Healthcare Surge Capacity during Pandemic Response"

_healthcare, 2023, doi:10.3390/healthcare11030324_

Round 1

Reviewer 1 Report

The manuscript is well written. The concept the evaluation of "Alternative Care Site Preparedness Assessment Tool" is relevant in current situation. Also, identification of need of revision of Review and revise the COVID-19 ACS PAT V2.0 checklist for ACS building selection is important for the need of advancement of current knowledge for the appropriate management of pandemic situation. 

specific comments:
The manuscript is well written. The concept the evaluation of "Alternative Care Site Preparedness Assessment Tool" is relevant in current situation. However, following points would be beneficial for further improvement of the manuscript.
Whole of society: Authors have mentioned about WHO model of whole of society. It is encouraged to discuss about collaboration and cooperation from the community as well. Community people’s awareness towards the particular pandemic problem might be the one hindering factor.
Alternative care sites: Authors have written well about alternative care sites used in different countries. But it would be better if authors discuss more about the most significant area to consider when developing alternative care sites.
Results:
Authors could explain a bit more about COVID-19 ACS PAT V1 so that the need of updated version V2 would be justifiable.
Conclusion:
Based on the review, it would be better if authors could suggest important focus for further improvement when revising ACS PAT V2. 

Author Response

Dear Reviewer 1:  Thank you for your thoughtful comments.  We will be uploading an adjusted manuscript with all text edits and the new Figure 5 with a statistical summary (frequency and percentage) of our findings.  Please note that below we have given Line numbers (e.g., 179 to 182) to locate specific text due to the number of changes.  The line numbers might be slightly off due to formatting with Word but you should be able to find them relatively easily in the range of our answers.  Thank you for your understanding of logistics in advance.  Below your questions are noted as:

Bolded text = reviewer question

Standard text = authors response

Specific comments:
The manuscript is well written. The concept the evaluation of "Alternative Care Site Preparedness Assessment Tool" is relevant in current situation. However, following points would be beneficial for further improvement of the manuscript.

1) Whole of society: Authors have mentioned about WHO model of whole of society. It is encouraged to discuss about collaboration and cooperation from the community as well. Community people’s awareness towards the particular pandemic problem might be the one hindering factor.

Thank you for your comment.  We read additional works [Schwartz et al 2017, Epps et a. 2021, and The Independent Panel for Pandemic Preparedness and Response background paper #10 Centering Communities (appointed by WHO/WHA)] and inserted text related to community engagement and collaboration.  See Sections:

2.3 Whole-of Society Response & Built Environment

Lines 130 – 132 Text inserted.

This includes community engagement throughout the pandemic cycle to build trust, share resources, and establish consensus decision making for increased compliance to public health control measures.

2.6 US COVID-19 Developing Situation

Lines 231-236 Text Inserted:

Introducing statements of uncertainty during a pandemic can create fear, mistrust, and panic.  This absence of clarity and trust can lead to a lack of community engagement and ultimately dissuade persons from compliance with public health measures at the local level. This can lead to a lack of community engagement and ultimately dissuade persons from compliance with public health measures at the local level. To promote effective engagement, it is necessary to identify sources of reliable information and avoid spreading rumors.

2) Alternative care sites: Authors have written well about alternative care sites used in different countries. But it would be better if authors discuss more about the most significant area to consider when developing alternative care sites.

Thank you for your comment.  We have added text in Results, Discussion, and Conclusions to emphasize the most significant areas. See responses in #3 below.  

3) Results: Authors could explain a bit more about COVID-19 ACS PAT V1 so that the need of updated version V2 would be justifiable.

Thank you for your comment.  In specific response to your suggestion, we have added the following information to improve our results and justify an update:

  1. Adjusted Results section description related to Article search and coding results (see lines 336 – 369).
  2. Updated the Results section by adding Figure 5 (See lines 404, p.11 for new illustrated chart) with summary statistics to identify frequency and percentage of responses to evaluate alignment of topics between the ACS PAT checklist and the reported items from the ACS case studies.
  3. Added text to the Discussion sections when appropriate to insert statistical frequency and percentages (example: n=11, 47.8%) to indicate the degree of alignment between COVID-19 ACS PAT checklist and ACS Case Studies.  Test was altered for this on lines:  390, 432 – 437, 472, 487, 504, 525, 533-534, 577-582, 585, 656, 692, 698, 706, 720, 731, 740, 758, and 760.  

4) Conclusion: Based on the review, it would be better if authors could suggest important focus for further improvement when revising ACS PAT V2.

Thank you for your comment.  We have better summarized our findings and recommendations in the conclusion as to which COVID-19 ACS PAT sections and subtopics need to be updated. See lines 849 – 895 for revised text. 

Reviewer 2 Report

Thank you for the opportunity to review this outstanding work. This paper will contribute to strengthening disaster preparedness in the healthcare system, particularly for pandemic disasters that have specific requirements. The introduction provides sufficient information about the holistic approach to disasters related to pandemics and offers a brief glimpse into the principles of disaster management related to the healthcare system. However, I suggest adding more details on COVID-19’s negative impacts on hospitals in terms of staffing, space, and supplies. Additionally, it would be helpful to add efforts from US authors to the introduction and discussion sections, such as the following:

·      Orsini E., Mireles-Cabodevila E., Ashton R., Khouli H., and Chaisson N. Lessons on outbreak preparedness from the Cleveland Clinic. Chest. 2020, Nov 1, 158(5):2090–6.

·      Meyer, D., Martin, E.K., Madad, S., Dhagat, P., and Nuzzo, J.B. Preparedness and response to an emerging health threat—Lessons learned from Candida auris outbreaks in the United States. Infection Control & Hospital Epidemiology. 2021, 42(11): 1301–1306.

·      Schultz, C.H., Koenig, K.L., Whiteside, M., Murray, R., and National Standardized All-Hazard Disaster Core Competencies Task Force. Development of national standardized all-hazard disaster core competencies for acute care physicians, nurses, and EMS professionals. Annals of Emergency Medicine. 2012, 59(3): 196–208.

Thanks again.

Author Response

Dear Reviewer 2:  Thank you for your thoughtful comments.  We will be uploading an adjusted manuscript with all responses.  Please note that below we have given Line numbers (e.g., 179 to 182) to locate specific text due to the number of changes.  The line numbers might be slightly off due to formatting with Word but you should be able to find them relatively easily in the range of our answers.  Thank you for your understanding of logistics in advance.  Below your questions are noted as:

Bolded text = reviewer question

Standard text = authors response

1) Checked Box for “English very difficult to understand / incomprehensible”:

Thank you for you evaluation, yet we assume this box may have been check in error.  You note the work is “outstanding” and therefore we assume you can read the manuscript proficiently to make such an assessment.  The current checked box (English very difficult to understand / incomprehensible) is leading the journal editorial staff to think we have to hire someone who speaks native English to rewrite the manuscript, which respectfully is not the case.  We are native English-speaking writers with extensive experience in publication for scientific manuscripts involving the built environment. We respectfully ask you to change the English writing evaluation box marking to a more appropriate English editing as only needs spellcheck and punctuation.  If you still feel the language is difficult to understand we would need specific examples to address your concerns. 

2) I suggest adding more details on COVID-19’s negative impacts on hospitals in terms of staffing, space, and supplies. Additionally, it would be helpful to add efforts from US authors to the introduction and discussion sections, such as the following:

  • Orsini E., Mireles-Cabodevila E., Ashton R., Khouli H., and Chaisson N. Lessons on outbreak preparedness from the Cleveland Clinic. Chest. 2020, Nov 1, 158(5):2090–6.
  • Meyer, D., Martin, E.K., Madad, S., Dhagat, P., and Nuzzo, J.B. Preparedness and response to an emerging health threat—Lessons learned from Candida auris outbreaks in the United States. Infection Control & Hospital Epidemiology. 2021, 42(11): 1301–1306.
  • Schultz, C.H., Koenig, K.L., Whiteside, M., Murray, R., and National Standardized All-Hazard Disaster Core Competencies Task Force. Development of national standardized all-hazard disaster core competencies for acute care physicians, nurses, and EMS professionals. Annals of Emergency Medicine. 2012, 59(3): 196–208.

Thank you for your comment and these articles.  We read each of these articles, added text, and citations throughout the manuscript.  The primary text added can be reviewed from Lines 182 to 196. We also used these citations to refer to all-hazards competency skills and training as part of our healthcare design professional analysis (see lines 577 to 582) and is also part of our recommendations. 

Reviewer 3 Report

This paper is a very interesting description of the work done by the Alternative Care Sites (ACS) Working Group of the American Institute of Architects (AIA) during the COVID-19 pandemic, however, the paper lacks scientific rigour.

The authors have described a process of the work and provided a very interesting review of the literature, but the results do not answer any research question.

The objective is ambiguous: what is the research question? what is to be demonstrated?

Undoubtedly, the topic of this paper is very relevant and deserves to be treated as a research topic, but for that a more concrete objective should be proposed.

The conclusions do not bring anything new to the scientific community.

The authors are invited to investigate in this line of work, however, they should restructure the paper because it is not considered suitable for publication in its current state.

Author Response

Dear Reviewer 3:  Thank you for your thoughtful comments.  We will be uploading an adjusted manuscript with all responses and the new Figure 5 with a statistical summary (frequency and percentage) of our findings.  Please note that below we have given Line numbers (e.g., 179 to 182) to locate specific text due to the number of changes.  The line numbers might be slightly off due to formatting with Word but you should be able to find them relatively easily in the range of our answers.  Thank you for your understanding of logistics in advance.  Below your questions are noted as:

Bolded text = reviewer question

Standard text = authors response

1) This paper is a very interesting description of the work done by the Alternative Care Sites (ACS) Working Group of the American Institute of Architects (AIA) during the COVID-19 pandemic, however, the paper lacks scientific rigor.

Thank you for your comment.  Respectfully, we have interpreted your collective comments as being supportive of the research topic but dissatisfied with the lack of clarity to evaluate the ACS PAT checklist when compared to the ACS published case studies.  We initially described research aims at the end of the introduction in a broad context intending that to suffice as the research question(s).  We stated the study’s purpose as broad research aims since we were not examining traditional variable relationships (dependent vs independent variables, p values, or other traditional quantitative methods of analysis).  To bring more clarity and basic statistical summary to evaluate the criteria stated in the ACS PAT checklist we:

  1. Added a third research aim (see line 80).
  2. Added one research question and three clarifying research objectives (see lines 249 to 256).
  3. Performed summary statistical analysis (frequency and percentage) and created Figure 5 to provide the reader with the ability to compare key sections and subtopics between the ACS PAT checklist and the ACS published case studies (see Figure 5 line 404).
  4. Updated text throughout the manuscript where necessary to insert the statistical analysis to clarify what we are referring to as alignment between the ACS PAT checklist and the ACS published case studies on how to evaluate, select and modify the built environment for safe patient care operations in buildings that were never intended for such purpose. Please review the edited manuscript; the primary text modifications related to this comment can be found on lines:  345 – 369, Figure 5 on line 404 (p. 11), and throughout the discussion section.

2) The authors have described a process of the work and provided a very interesting review of the literature, but the results do not answer any research question.

Thank you for your comment.  See answer provided under Item 1 above.

3) The objective is ambiguous: what is the research question? what is to be demonstrated?

Thank you for your comment.  See answer provided under Item 1 above.

4) Undoubtedly, the topic of this paper is very deserves to be treated as a research topic, but for that a more concrete objective should be proposed.

Thank you for your comment.  See answer provided under Item 1 above.

5) The conclusions do not bring anything new to the scientific community.

Thank you for your comment. Respectfully, we have taken your collective comments in addition to this statement that you were looking for more definitive recommendations and conclusions.  We have updated the conclusion section based on our additional findings from the clarified research question and objectives.  We rewrote the conclusion section (see lines 849 – 895). 

With respect to contributing to the scientific community, it has been previously reported in numerous peer-reviewed journals that the built environment (both related and unrelated to pandemic response) has contributed to unnecessary patient illness, injury, and death.  In this study, we are now reporting our experience for creating guidance for healthcare professionals and other stakeholders who were forced to manage healthcare facilities and operations in non-healthcare buildings with little to no direction during the COVID-19 pandemic, as well as synthesizing the findings of numerous other studies which offered their own narrative descriptions about their lessons learned. We believe that summarizing the collective learning(s) in the context of the rapid response development of a guidance tool will result in a safer built environment to conduct healthcare operations outside of a traditional hospital setting during disaster events -- an important topic to address in public health preparedness.

6) The authors are invited to investigate in this line of work, however, they should restructure the paper because it is not considered suitable for publication in its current state.

Thank you for your comment.  As stated prior we have added significantly to the manuscript to address what we perceive as your concerns around a research question and quantitative analysis.  We hope we have interpreted your concerns to your satisfaction.